# ADCK2 Knockdown Affects the Migration of Melanoma Cells via MYL6

**DOI:** 10.3390/cancers14041071

**Published:** 2022-02-20

**Authors:** Marlene Vierthaler, Qian Sun, Yiman Wang, Tamara Steinfass, Juliane Poelchen, Thomas Hielscher, Daniel Novak, Viktor Umansky, Jochen Utikal

**Affiliations:** 1Skin Cancer Unit, German Cancer Research Center (DKFZ), 69120 Heidelberg, Germany; m.vierthaler@dkfz.de (M.V.); q.sun@dkfz.de (Q.S.); yiman.wang@dkfz.de (Y.W.); t.steinfass@dkfz.de (T.S.); j.poelchen@dkfz.de (J.P.); d.novak@dkfz.de (D.N.); v.umansky@dkfz.de (V.U.); 2Department of Dermatology, Venereology and Allergology, University Medical Center Mannheim, Ruprecht Karl University of Heidelberg, 68167 Mannheim, Germany; 3DKFZ-Hector Cancer Institute, University Medical Centre Mannheim, 68167 Mannheim, Germany; 4Faculty of Biosciences, Ruprecht Karl University, 69120 Heidelberg, Germany; 5Biostatistik, German Cancer Research Center (DKFZ), 69120 Heidelberg, Germany; t.hielscher@dkfz.de

**Keywords:** melanoma, ADCK2, MYL6, motility, cancer

## Abstract

**Simple Summary:**

Melanoma is a growing health issue in the 21st century. Due to early metastasis and the development of resistance, it still goes along with a poor prognosis. ADCK protein kinases have been shown to play a role during cancer development and metastasis. Here, we investigated the role of ADCK2 in melanoma. In our study, we showed that higher levels of intratumoral ADCK2 benefit patient survival, while a low expression of ADCK2 was associated with a higher motility and a dedifferentiated state of melanoma cells, which facilitates metastasis. Our results could give new insights into melanoma metastasis, and ADCK2 could qualify as a prognostic marker or a target for melanoma therapy in the future.

**Abstract:**

Background: ADCK2 is a member of the AarF domain-containing kinase family, which consists of five members, and has been shown to play a role in CoQ metabolism. However, ADCKs have also been connected to cancer cell survival, proliferation and motility. In this study, we investigated the role of ADCK2 in melanoma. Methods: The effect of ADCK2 on melanoma cell motility was evaluated by a scratch assay and a transwell invasion assay upon siRNA-mediated knockdown or stable overexpression of ADCK2. Results: We found that high levels of intratumoral ADCK2 and MYL6 are associated with a higher survival rate in melanoma patients. Knocking down ADCK2 resulted in enhanced cell migration of melanoma cells. Moreover, ADCK2-knockdown cells adopted a more dedifferentiated phenotype. A gene expression array revealed that the expression of ADCK2 correlated with the expressions of MYL6 and RAB2A. Knocking down MYL6 in ADCK2-overexpressing cells could abrogate the effect of ADCK2 overexpression and thus confirm the functional connection between ADCK2 and MYL6. Conclusion: ADCK2 affects melanoma cell motility, most probably via MYL6. Our results allow the conclusion that ADCK2 could act as a tumor suppressor in melanoma.

## 1. Introduction

Our skin is the first barrier to shield the body from potentially harmful external influences, thereby protecting us from, e.g., ultraviolet (UV) light, bacterial infections, and chemical toxins. A breakdown of this barrier often leads to severe skin diseases, whose incidences increased over the past years [1]. One of these diseases is melanoma, which is an aggressive form of skin cancer with about 55,500 deaths per year worldwide [2,3]. Although there are many therapeutic options, such as surgery, targeted therapy and immunotherapy, the disease is still very often associated with a fatal outcome. This is mainly due to early metastatic spread and the development of resistance [4,5]. That is why it is of great importance to learn more about the development and pathogenesis of melanoma.

The ADCK (AarF domain-containing kinase) kinases belong to the atypical kinase family and were first described in 1998 [6,7]. There are five members (ADCK1-5) that all carry a highly conserved aarF domain [6,8]. Research on ADCK members (ADCK2, 3 and 4) has shown their involvement in ß-oxidation and CoQ (coenzyme Q) metabolism [9,10,11,12]. Furthermore, some ADCKs have been reported to play a role in cancer. Egashira and colleagues demonstrated the occurrence of ADCK4-NUMBL (ADCK4-Numb-like protein) fusion proteins in cancers on sun-exposed skin areas [13]. ADCK5 is upregulated in many cancers, and upregulation of ADCK2 promotes the survival of luminal breast cancer cells [14,15]. This positive effect of ADCK2 on the viability was also reported for other ER+ breast cancer cell lines and also for glioblastoma multiforme cells [8,16]. Furthermore, ADCK2 expression correlates with cancer cell proliferation and motility and a higher expression in breast cancer went along with a better tumor shrinkage after long-term NAET (Nambudripad’s Allergy Elimination Technique) treatment [10,14].

Myosins are motor proteins that interact with actin filaments and exert movement by hydrolyzing ATP (adenosine triphosphate). For this reason, myosins play an important role in several processes, including muscle contraction, cytokinesis, intracellular trafficking, signal transduction and cell movement [17]. Myosins are hexamers that consist of two heavy chains, harboring a conserved head domain, a neck domain and a C-terminal tail. The neck domain allows binding two regulatory light chains (RLCs), which can be phosphorylated and thereby activate or inactivate the whole myosin molecule and two essential light chains (ELCs) that most probably help stabilize the myosin structure [17,18,19]. Myosin light chain 6 (MYL6) is an ELC, which is bound by myosin heavy chain 14 (MYH14) and smooth muscle myosin. The complex of MYH14, MYL6 and RLCs (MYL9, 12A or 12B) is called non-muscle myosin 2C (NM2C) [17,20]. NM2s are important for the assembly of cytoskeletal structures.

In this study, we examined the role of ADCK2 in melanoma. With a knockdown approach, we demonstrated that ADCK2 has a slight effect on cell viability but a great effect on the migration of several melanoma cell lines. A gene expression array confirmed a positive correlation between ADCK2 expression and RAB2A (Ras-related protein Rab-2A) and MYL6 expression in different melanoma cell lines. Additionally, we found that the effect of ADCK2 on migration was mediated by MYL6, possibly due to its contribution to non-muscle myosin 2C (NM2C).

## 2. Materials and Methods

### 2.1. Cell Culture

All melanoma cell lines and HEK 293T cells were cultured in MEF medium (DMEM + GlutaMAXX (Gibco, Darmstadt, Germany) supplemented with 10% FCS (Biochrom, Berlin, Germany), 1% penicillin/streptomycin (Sigma-Aldrich, Steinheim, Germany), 1% non-essential amino acids (Sigma-Aldrich, Steinheim, Germany) and 0.1 mM ß-mercaptoethanol (Gibco, Darmstadt, Germany) and were regularly split with 0.05% trypsin EDTA (Gibco, Darmstadt, Germany) when they reached full confluency. All cells were grown at 37 °C and 5% CO_2_.

### 2.2. siRNA Transfection and Overexpression

Cells were seeded in normal MEF medium one day prior to transfection/infection. The siRNA transfection was conducted with Lipofectamine RNAi MAXX (Thermo Fisher Scientific, Schwerte, Germany) and was implemented according to the manufacturer’s protocol with slight changes. In brief, Lipofectamine RNAi MAXX was mixed with DMEM/F-12 (Gibco, Darmstadt, Germany) in one tube and siRNAs (3 µL of 10 µM stock solution per well of a 6-Well plate) were mixed with DMEM/F-12 in a second tube. Then, the Lipofectamine mix was transferred to a fresh tube and was mixed with siRNA mix by carefully pipetting up and down. Before the Lipofectamine siRNA mix was added dropwise to the cells, the medium was changed to MEF medium. Cells were incubated at 37 °C and 5% CO_2_ for different time periods. All siRNAs were obtained from Qiagen as follows: AllStars Neg. siRNA AF 488, Hs_ADCK2_5 FlexiTube siRNA, Hs_MYL6_3 FlexiTube siRNA.

For the production of viral particles carrying the ADCK2 overexpression construct, HEK293T cells were transfected with X-tremeGENE 9 DNA Transfection Reagent (Roche, Mannheim, Germany) according to the manufacturer´s protocol. In brief, cells were seeded and grown in MEF medium until they reached about 70% confluency. The X-treme GENE 9 DNA Transfection reagent was added to DMEM (Gibco, Darmstadt, Germany) without supplements, and the mix was incubated for 5 min at RT. Then, two packaging plasmids (5.5 µg pCMV-VSV G, 8.25 µg pCMV-Δ8.9) and 11 µg of the ADCK2 OE vector were added and incubated for 30 min at RT. Fresh MEF medium was given to the cells, and then the transfection reagent-DNA mix was added dropwise to the medium. Afterwards, cells were incubated at 37 °C and 5% CO_2_ overnight. The next day, the medium was discarded, and a fresh MEF medium was added. Medium containing viral particles was collected three times every 12 h. For infection of melanoma cells, SkMel28, MeWo, A375 and SkMel30 cells were seeded and grown until they reached about 70% confluency. Sterile-filtered virus and 5 µg/mL polybrene (Sigma-Aldrich, Steinheim, Germany) were added, and cells were incubated at 37 °C and 5% CO_2_ for 24 h, followed by a second infection without polybrene for 24 h. Next, cells were thoroughly washed with PBS (Gibco, Darmstadt, Germany), and a fresh MEF medium was added. Successfully transduced cells were selected with 0.5 µg/mL puromycin (Carl Roth, Karlsruhe, Germany). The ADCK2 overexpression vector was obtained from VectorBuilder (Neu-Isenburg, Germany) (pLV[Exp]-Puro-EF1A > hADCK2[NM_052853.4] (Vector ID: VB900090-4328zyg)). The empty vector control was generated by removing the open reading of ADCK2 from the overexpression construct.

### 2.3. qPCR

RNA was extracted from cell pellets with the Qiagen (Hilden, Germany) RNeasy Kit according to the manufacturer´s protocol, including all optional steps. 500 µg of RNA were then transcribed into cDNA using the RevertAid First Strand cDNA Synthesis Kit from Thermo Fisher Scientific (Schwerte, Germany)according to the manufacturer´s protocol, using Oligo dT primer. To determine expression levels, the QuantiNova SYBR Green Kit from Qiagen (Hilden, Germany)was used according to the manufacturer´s protocol with small changes regarding the composition of the MasterMix: 5 µL SybR Green, 0.05 µL ROX Reference Dye, 2.3 µL H_2_O and 0.15 µL primer working solution. 18S served as internal control.

All primers were validated and showed an efficiency between 90 and 110%. The sequences can be found in Appendix A.

### 2.4. Western Blot

Proteins were isolated by resuspending cell pellets in RIPA buffer (Sigma-Aldrich, Steinheim, Germany) containing protease inhibitors (cOmplete Mini tablets; Roche, Mannheim, Germany) and incubation for 30 min on ice. Afterwards, the mix was centrifuged at maximum speed for 30 min at 4 °C. The supernatant was transferred to a fresh tube, and the protein concentration was measured with a BCA assay (Pierce BCA Protein Assay Kit, Thermo Fisher Scientific, Schwerte, Germany). For Western blot, NuPAGE Bis Tris MiniGels (4–12%) from Invitrogen (Darmstadt, Germany) were used, and 30 µg of protein per sample was loaded. After running at 200 V for about 45 min, proteins were transferred to an Immobilon P membrane (Merck, Darmstadt, Germany) at 100 V for 1 h. Afterwards, the membrane was blocked with 5% milk (Skim Milk Powder, GERBU, Heidelberg, Germany) or BSA (Albumin Fraction V, Carl Roth, Karlsruhe, Germany) for 1 h at RT, followed by incubation with Anti-ADCK2 (ab235312 from Abcam, Cambridge, UK), Anti-MYL6 Polyclonal (Invitrogen, Darmstadt, Germany) or Anti-GAPDH (ab14C10 from Cell Signaling Technology, Frankfurt am Main, Germany) antibody overnight at 4 °C. After several washing steps with TBS-T, the membrane was incubated with anti-rb IgG HRP-linked antibody (Cell Signaling Technology, Frankfurt am Main, Germany) for 1h at RT, followed by several washing steps with TBS-T. Specific bands were detected with the Immobilon Forte Western HRP Substrate (Merck, Darmstadt, Germany) and the ChemiDoc Touch Imaging System (BioRAD, Munich, Germany). The bands were analyzed with Fiji software (ImageJ), and GAPDH was used as loading control. This quantitative analysis was implemented in the figures by the numbers underneath the Western blot pictures. The supplementary information section also contains the original, unmodified Western blot images.

### 2.5. Immunofluorescence Staining (IF)

For IF, SkMel28 cells were seeded at about 60% confluency onto glass cover slides (neolab, Heidelberg, Germany) and transfected with either siADCK2 or siControl. After 48 h of incubation at 37 °C and 5% CO_2_, the medium was changed, and cells were further incubated at 37 °C and 5% CO_2_ for 12 h. Then, cells were washed with PBS (Gibco, Darmstadt, Germany), fixed with 4% paraformaldehyde (Sigma-Aldrich, Steinheim, Germany) and permeabilized with 0.1% tritonX-100 (Carl Roth, Karlsruhe, Germany). Afterwards, cells were blocked with 3% BSA (Carl Roth, Karlsruhe, Germany) and 0.3% tritonX-100 and incubated with anti-MYL6 polyclonal antibody (Invitrogen, Darmstadt, Germany) over night at 4 °C. The next day, cells were washed and incubated with an anti-rabbit IgG antibody coupled with Alexa Fluor 488 (Invitrogen, Darmstadt, Germany) at RT for 1 h. Afterwards, cells were washed and incubated simultaneously with ActinRed 555 (Life Technologies, Darmstadt, Germany) and DAPI (Roche, Mannheim, Germany) for 20 min at RT. Next, cells were mounted (Fluoromount Aqueous Mounting Medium, Sigma-Aldrich, Steinheim, Germany) onto a microscope slide (R. Langenbrinck GmbH, Emmendingen, Germany). After drying overnight at RT, pictures were taken with a Zeiss (Oberkochen, Germany) 700 microscope.

### 2.6. Cell Viability Assay

To determine the cell viability of melanoma cells, they were seeded (SkMel28: 0.2 × 10^4^ cells/well, MeWo: 0.12 × 10^4^ cells/well, A375: 0.03 × 10^4^ cells/well, SkMel30: 0.19 × 10^4^ cells/well) in quintuples in a 96-well plate (Greiner, Kremsmünster, Austria) one day before transfection or first measurement and incubated at 37 °C and 5% CO_2_. The next day, they were transfected with siRNA as described above. After 24, 48, 72 and 96 h, 1:10 Alamar blue (Invitrogen, Darmstadt, Germany) was added and incubated at 37 °C and 5% CO_2_ for 4 h. Then, the absorbance at 535 nm and the emission at 590 nm were measured with the Tecan (Männedorf, Switzerland) infinite F200 PRO.

### 2.7. Migration Assay

The migration of melanoma cells was quantified with 2-well Culture-Inserts (ibidi, Gräfelfing, Germany). Cells were seeded (all numbers are cells per well of an insert: SkMel28: 1.68 × 10^4^, MeWo: 1.25 × 10^4^, A375: 0.8 × 10^4^, SkMel30: 1.87 × 10^4^, SkMel28 EV/OE ADCK2: 2.5 × 10^4^, MeWo: EV: 3.75 × 10^4^ OE ADCK2: 2.5 × 10^4^, A375 EV/OE ADCK2: 1.87 × 10^4^, SkMel30 EV/OE ADCK2: 3.28 × 10^4^) into the inserts so that they reached 100% confluency either the next day (EV/OE ADCK2 cells) or after 72 h (siRNA) and were incubated at 37 °C and 5% CO_2_. Cells were transfected with siRNA, as described above, the next day and incubated at 37 °C and 5% CO_2_ for 48 h. Then, inserts were removed, the cells were washed with PBS (Gibco, Darmstadt, Germany), and MEF medium containing 1:1000 aphidicolin (Sigma-Aldrich, Steinheim, Germany) was added. Pictures were taken with a Nikon (Düsseldorf, Germany) Eclipse Ti microscope until the gap was visually closed and then analyzed with TScratch software.

### 2.8. Invasion Assay

For the invasion assay, a transwell insert system (Cultrex Basement Membrane Extract Cell Invasion Assay, 24-well, R&D Systems, Minneapolis, MN, USA) was used according to the manufacturer´s protocol. In brief, the top chamber was coated with 0.2 × BME 4 h prior to cell seeding. Duplicates of SkMel28 EV and OE ADCK2 cells or SkMel28 cells treated prior with siControl or siADCK2 for 48 h were seeded (2.75 × 10^5^ cells/transwell) in FCS-free MEF medium in the top chamber and incubated at 37 °C and 5% CO_2_. As a chemoattractant, a complete MEF medium was used in the bottom chamber, which was changed after 48 h. After 96 h, cells in the top and bottom chamber were washed, and the cells in the bottom chamber were incubated for 1 h with calcein-AM dissociation solution. Invading cells in the bottom chamber were quantified by measuring the relative fluorescence units with an excitation at 485 nm and emission at 520 nm with a Tecan (Männedorf, Switzerland) infinite F200 PRO plate reader.

### 2.9. Gene Expression Analysis

Biological triplicates of high-quality RNA samples from melanoma cells (MeWo, SkMel30, C32) treated with siControl or siADCK2 for 24 h were sent to the Gene expression Core Facility at the DKFZ, and an Affymetrix (Santa Clara, California, USA) Clariom S human Chip was used for analysis. Affymetrix CEL files were RMA normalized and expression values log2-transformed. Differentially expressed probesets/genes between groups were identified using the empirical Bayes approach [21] based on moderated t-statistics, as implemented in the Bioconductor package limma [22]. *p*-values were adjusted for multiple testing using the Benjamini-Hochberg correction in order to control the false discovery rate. All analyses were performed with statistical software R 4.0 [23].

### 2.10. Statistical Analysis

The statistical analyses for all experiments were performed with at least three biological triplicates. The software GraphPadPrism was used for statistical analysis for all experiments, except the gene expression analysis. All graphs always show the mean with SEM.

## 3. Results

### 3.1. Patients with High Intratumoral ADCK2 Expression Have a Better Prognosis

In order to investigate the role of ADCK2 in melanoma, we checked the endogenous mRNA expression levels of nine melanoma cell lines (BRAF and NRAS WT: MeWo, BRAF-mutated: WM2664, A375, HT144, SkMel28, C32 and NRAS-mutated: SkMel30, SkMel103, SkMel173) and normal human melanocytes (NHM). We found that the mRNA expression level of ADCK2 did not depend on the mutational status of melanoma cells. Furthermore, we found a tendency for a slightly reduced expression of ADCK2 in melanoma cell lines compared to NHM, which, however, was not significant except for the cell line C32 (Figure 1A). Analyzing expression data from two datasets from the cBioPortal database (https://www.cbioportal.org/ accessed on 12 October 2021) revealed that a lower ADCK2 expression in melanomas correlates with a worse patient prognosis while a higher expression positively affects the survival (Figure 1B).

### 3.2. ADCK2 Is Important for the Cell Viability of Melanoma Cells

To examine the effect of ADCK2 on the cell viability of melanoma cells, siRNA-mediated knockdown and ectopic overexpression (OE) of ADCK2 were used. Cell viability was measured 24, 48, 72 and 96 h after seeding/transfecting the cells. Knocking down ADCK2 in SkMel28 cells led to a reduction in ADCK2 by about 50–60% on an mRNA level and was also validated on a protein level after 96 h (Figure 2A,B). (For the cell lines MeWo, A375 and SkMel30, see Appendix A.) This resulted in a reduction in cell viability of SkMel28 cells for all four time points by about 20% compared to cells that were transfected with a control siRNA (Figure 2E). Lower cell viability was also detected for MeWo and SkMel30 cells. For A375 cells, lower cell viability could be seen after 48, 72 and 96 h (Appendix A). Additionally, we stably overexpressed ADCK2 in SkMel28, A375, SkMel30 and MeWo cells with the help of a lentiviral expression construct in order to determine if this also affects the cell viability of melanoma cells. Ectopic overexpression of ADCK2 increased the mRNA level about three-fold for SkMel28, MeWo and SkMel30 and about five-fold for A375 cells compared to an empty vector (EV) control. This overexpression was also confirmed by Western blot (Figure 2C,D and Appendix A). The cell viability of SkMel28 OE ADCK2 was only slightly increased after 24 h whereas MeWo OE ADCK2, A375 OE ADCK2 and SkMel30 OE ADCK2 cells did not show any difference in cell viability (Figure 2F and Appendix A).

These results suggest that ADCK2 plays a role in melanoma cell viability and a lower expression of ADCK2 leads to lower cell viability.

### 3.3. ADCK2 Limits the Migration and Invasion of Melanoma Cells

Next, we assessed if ADCK2 also affects the migration and invasion capacity of melanoma cells. For this, a wound healing migration assay was performed. Forty-eight hours after knocking down ADCK2, the surface area of the open gap was measured at several defined time points. As can be seen in Figure 3A, the knockdown of ADCK2 led to a more migrative phenotype (smaller gap area) of SkMel28 cells compared to cells that were transfected with a control siRNA. A faster migration upon ADCK2 knockdown could also be detected for MeWo, SkMel30 and A375 cells (Appendix A). To verify the effect of ADCK2 on cell migration, the wound healing migration assay was repeated with ADCK2-overexpressing cells. The migration of SkMel28 OE ADCK2 cells was impaired compared to SkMel28 EV cells (Figure 3B).

We also examined the invasion capacity of SkMel28 cells upon siRNA-mediated KD of ADCK2 and in ADCK2-overexpressing SkMel28 cells by using the transwell invasion assay. Knocking down ADCK2 led to a significantly higher invasion capacity, and the overexpression of ADCK2 resulted in a significantly lower invasion capacity of SkMel28 cells (Figure 3C,D).

Our results show that ADCK2 has a considerable negative effect on migration and invasion of melanoma cells and, thus, might act as a tumor suppressor.

### 3.4. ADCK2 Affects the Differentiation Status of Melanoma Cells

We found a positive correlation between ADCK2 expression and the expression of some melanocyte markers, but a negative correlation between ADCK2 expression and the expression of the neural crest cell (NCC) marker p75 in SkMel28 and SkMel30 cells. The expression of the melanocyte marker TRP1 in SkMel28 cells decreased upon knockdown of ADCK2 compared to the control. The most significant downregulation was seen 72 and 96 h after knockdown. MITF expression was not significantly reduced, but we could observe a strong tendency for downregulation. In SkMel30 cells, TYR expression was significantly lower 96 h after the knockdown (Figure 4A,B). In contrast, p75 showed a higher mRNA expression in SkMel28 and SkMel30 cells after ADCK2 knockdown, especially after 72 and 96 h compared to SkMel28 and SkMel30 siControl samples. A similar trend for a decrease in melanocyte marker and an increase in NCC marker expression upon ADCK2 knockdown was also seen in MeWo and A375 cells (Appendix A). Furthermore, cell pellets of SkMel30 cells were slightly brighter after transfection with siADCK2, indicating a reduction in pigment synthesis. Conversely, pellets of ADCK2-overexpressing cells were much darker, suggesting an increased amount of pigment (Figure 4C). These results show that ADCK2 influences the differentiation status of melanoma cells.

### 3.5. MYL6 Is Functionally Connected to ADCK2

Next, we wanted to identify effectors that are up- or downstream of ADCK2. For this purpose, we performed a gene expression array with three melanoma cell lines (MeWo, SkMel30, C32). Samples were taken 24 h after knockdown of ADCK2 with siADCK2 and then compared to cells transfected with siControl. Gene expression analysis with regard to genes involved in cell migration revealed considerable differences between siControl- and siADCK2-transfected cells (Appendix A). Comparing all significantly up- and downregulated genes in MeWo, SkMel30 and C32 cells revealed four common downregulated genes (Appendix A), namely ADCK2, RAB2A, ZNF275 (zinc finger protein 275) and MYL6, confirming the successful knockdown of ADCK2. RAB2A is a known oncogene in many cancers but has not yet been studied in melanoma. ZNF275 is a protein harboring a zinc-finger motif that is most probably involved in DNA binding. Nothing is known about its function so far. Of utmost interest for us was MYL6, which is an ELC of myosin and, as such, is involved in controlling cell motility. 

An mRNA expression analysis revealed a lower expression of MYL6 in SkMel28 cells 24, 48, 72 and 96 h after ADCK2 knockdown. The expression was significantly reduced by at least 40% and up to 80% (Figure 5A) in SkMel28 cells. A similar result was observed for SkMel30, MeWo and A375 cells (Appendix A). Immunofluorescence staining confirmed the lower expression of MYL6 upon ADCK2 knockdown in SkMel28 cells (Figure 5B). Furthermore, the R2: Genomics analysis and visualization platform (https://r2.amc.nl (Access on 12 October 2021)) also confirmed a positive correlation between ADCK2 and MYL6 expression in two distinct melanoma sample sets (Figure 5C). Data from the same database as well as from the cBioPortal database (https://www.cbioportal.org/ (accessed on 12 October 2021) indicate that a lower intratumoral expression of MYL6 is correlated with a poorer survival outcome (Figure 5D). 

### 3.6. Knockdown of MYL6 Negates the Effects of ADCK2 Overexpression on Melanoma Cells

Next, we were interested if the positive correlation between ADCK2 and MYL6 expression might go along with a functional connection. First, we demonstrated that the knockdown of MYL6 did not affect the expression of ADCK2 (Figure 6A), indicating that ADCK2 might be upstream of MYL6. In order to investigate a putative functional connection between these two proteins, we overexpressed ADCK2 while simultaneously knocking down MYL6 with a specific siRNA. A successful knockdown of MYL6 of about 80 % in SkMel28 OE ADCK2 cells was confirmed 24, 48, 72 and 96 h after transfection (Figure 6C). A successful knockdown of MYL6 was also seen in MeWo OE ADCK2 cells (Appendix A). This knockdown slightly decreased the cell viability of both tested ADCK2-overexpressing cell lines compared to the siControl transfection (Figure 6D and Appendix A). Since knocking down ADCK2 significantly altered the migration capacity of melanoma cells, we also performed a wound healing migration assay after knocking down MYL6 in OE ADCK2 cells. We measured increased migration of SkMel28 OE ADCK2 and MeWo OE ADCK2 cells after MYL6 knockdown compared to siControl-transfected cells (Figure 6E and Appendix A). Interestingly, pellets from SkMel30 OE ADCK2 cells transfected with siMYL6 were brighter than the ones from SkMel30 cells transfected with siControl (Appendix A). This observation was in line with our previous finding that knocking down ADCK2 resulted in decreased pigmentation (Figure 4C).

With these experiments, we confirmed a functional connection between ADCK2 and MYL6. Our observation that the overexpression of ADCK2 did not alter the expression of MYL6 (Figure 6B), however, suggests that ADCK2 does not affect the function of MYL6 directly.

## 4. Discussion

In this study, we could demonstrate that ADCK2 affects different properties of melanoma cells and that these effects are mediated via MYL6. Analyses of publicly available data sets revealed that patients with high intratumoral ADCK2 expression have a survival benefit over patients with low intratumoral ADCK2 expression. As opposed to this, it was shown in ER+ (estrogen receptor-positive) /HER-negative (human epidermal growth factor receptor 2-negative) breast cancer that ADCK2 overexpression correlated with a worse outcome (abstract from AACR (cancerres.aacrjournals.org/content/80/4_Supplement/P2-11-17, accessed on 12 October 2021)). Therefore, it seems that ADCK2 plays different roles in distinct cancer types. 

Next, we assessed the effect of ADCK2 on melanoma cell viability. We could observe that the viability of SkMel28 cells decreased by about 20% upon ADCK2 knockdown. Lower viability after knocking down ADCK2 could also be confirmed in several cell lines from breast, lung and cervical cancer [14] and is assumed for retinoblastoma [16]. In ER+ breast cancer cells, the effect of ADCK2 on survival is mediated by its interaction with the estrogen receptor ERalpha and thereby altered estrogen signaling [8]. In osteosarcoma and prostate cancer cells, ADCK2 was found to promote the TNFα-mediated accumulation of HIF-1α stability, which, in turn, is associated with the proliferation and survival of cancer cells. Moreover, ADCK2 influences this mechanism via non-canonical NFκB signaling [24]. The exact mechanism of how ADCK2 is influencing the viability of melanoma cells needs to be further investigated.

In our study, we were able to show that ADCK2 had a strong effect on the motility of melanoma cells. Cell motility plays an important role in various cellular processes, including tissue remodeling, development, tumorigenesis and metastasis. All these processes are controlled by a complex network, which involves many different genes contributing directly or indirectly to cell motility [25]. A knockdown of ADCK2 clearly increased the migrative capacity of SkMel28, A375, SkMel30 and MeWo cells. Furthermore, a significantly lower migrative and invasive capacity of ADCK2-overexpressing SkMel28 cells was detected. Several members of the ADCK family have already been shown to play a role in migration. ADCK5 phosphorylates Sox9 and thereby regulates PTTG1 in lung cancer cells, which leads to a change in the migrative and invasive behavior of these cells [15]. In epithelial cells, ADCK1 and ADCK4 have been connected to cell migration; in particular, a faster migration was seen upon the knockdown of ADCK4 [25], and locomotor dysfunctions have been observed in heterozygous ADCK2 knockout mice [12]. A higher migration and invasion rate of cancer cells is usually associated with the ability to form metastases. Our results show that a reduction in ADCK2 expression rendered melanoma cells more migrative and invasive. Montagnani and colleagues predicted ADCK2 as a metastasis-associated gene in melanoma, especially in BRAF-mutated melanoma [26]. In addition, a fusion of BRAF and ADCK2 has been detected in infantile fibrosarcoma [27]. To further analyze the more migrative and invasive phenotype of melanoma cells upon the knockdown of ADCK2, the mRNA expression of different melanocyte and NCC markers was quantified. We observed a significantly lower expression of the melanocyte markers TYR and TRP1 and a higher expression of the NCC marker p75, which leads to the assumption that a knockdown of ADCK2 led to a dedifferentiation of melanoma cells, prompting them to adopt a more aggressive phenotype. This assumption was supported by the finding that the pigmentation of melanoma cells was reduced upon ADCK2 knockdown. TYR is also a marker for a proliferative phenotype of cells, whereas Wnt5a, TGFβ and FGF are markers for an invasive phenotype [28]. The downregulation of TYR and TRP1 in siADCK2-treated melanoma cells confirmed the switch to a more migrative and invasive behavior. In order to unravel the mechanism of how ADCK2 altered the migrative capacity of melanoma cells, we performed a gene expression analysis of three melanoma cell lines upon ADCK2 knockdown. We found a positive correlation between ADCK2, RAB2A and MYL6 expression on RNA level. MYL6, which is an ECL within the NM2 complex and therefore involved in regulating the actin cytoskeleton, was examined in more detail. The function of ECLs is not yet clear, but they most likely play a role in stabilizing the NM2 complex. NM2s contribute to cell motility, and aberrant expression was reported in different diseases, including cancer. In humans, there are three different forms of NM2s (NM2A, NM2B and NM2C), and MYL6 was found to bind to MYH14 and RLC9, 12A or 12B to form NM2C [19,29]. The positive correlation that we detected between ADCK2 and MYL6 expression was validated on RNA and protein levels and could be also confirmed by an examination of data from the GEO database. Moreover, analyzing additional data from the GEO and the cBioPortal database revealed that low levels of intratumoral MYL6 correlate with poor prognoses of melanoma patients. In lung carcinoma, a machine learning model could associate MYL6 expression with an “alive without cancer” group, and in a neuroblastoma patient cohort, a higher intratumoral expression of MYL6 promised a better outcome [30,31]. To further investigate if the effect of ADCK2 on viability and migration was mediated via MYL6, we ectopically overexpressed ADCK2 while simultaneously knocking down MYL6 in melanoma cell lines. Interestingly, the knockdown of MYL6 counteracted the effect of ADCK2 overexpression on cell viability and migration capacity. Various studies connected decreased levels of NM2 variants with a higher migration rate. Reduced expression of NM2A results in a faster migration but lower cell contractility, which is most likely due to increased microtubule stability [29]. Furthermore, lower levels of NM2A are associated with tumor development and metastasis of skin cancer [19]. In squamous cell carcinoma, a reduced expression level of MYH9 (the heavy chain of NM2A) was associated with a higher invasion rate and distant metastases. Additionally, here, a lower expression level of MYH9 led to a poorer survival rate [32]. In several studies, it has been reported that a shift of the myosin dynamics to disassembling NM2A or NM2B correlates with an increase in cell migration and invasion due to less condensed actin structures [19,33]. siRNA-mediated knockdown of the RLCs MYL12A and 12B in NIH 3T3 fibroblasts resulted in a lower expression of MYH9 (the heavy chain of NM2A), MYH10 (the heavy chain of NM2B) and MYL6. These cells also showed an altered morphology with protrusions, a defective formation of actin fibers and reduced cell contractility and induced migration [34]. In contrast to these observations, it has also been shown that NM2 activity can have a promoting effect on cancer cell invasion and metastasis. In esophageal cancer, for example, NM2A overexpression correlates with more lymph node metastases, poor cancer differentiation and advanced tumor stages [29]. Taking all these studies into account, it seems that NM2s can have tumor-suppressive and tumor-promoting effects depending on the cellular background. So far, it remains unclear how a reduction in MYL6 enhances cancer cell migration. Lower expression of MYL6 could reduce the stability of RLCs and MYH chains, resulting in the disassembly of NM2s, which, in turn, entails a deformation of actin filaments and hence more migration. Alternatively, it could also be possible that with the absence of MYL6, a regulatory influence on the MYH chains is lost, resulting in a constitutively active form of NM2s. This could lead to more activation of NM2s on actin filaments and could thereby also enhance the migration of melanoma cells. It would be interesting to examine if altered MYL6 levels directly influence the protein levels of other ELCs or RLCs and if MYL6 knockdown significantly alters the architecture of the actin cytoskeleton. In any case, additional studies are necessary to clarify how MYL6 controls cell migration. 

Our observation that a knockdown of ADCK2 led to a lower expression of MYL6 and shifted melanoma cells to a more dedifferentiated and aggressive phenotype is in line with the above-cited research findings.

## 5. Conclusions

In this study, we showed that altering the expression of ADCK2 affected the cell viability and the migrative and invasive capacity of melanoma cells. Knocking down ADCK2 reduced the cell viability and simultaneously increased the migration, most likely via MYL6. The opposite effect was observed with ADCK2 overexpression, which resulted in decreased migration invasion capacity. The fact that a lower intratumoral expression of ADCK2, as well as MYL6, in melanoma is connected with a poor prognosis for melanoma patients allows the conclusion that ADCK2 might be a tumor suppressor in melanoma. ADCK2 could control the migration of melanoma cells by affecting the formation of NM2s via MYL6. Furthermore, we demonstrated that a knockdown of ADCK2 led to a more aggressive phenotype of melanoma cells, suggesting that ADCK2 might be a suitable prognostic marker. Understanding the consequences of the downregulation of ADCK2 in melanoma cells could additionally open up new possibilities for therapeutic approaches.

## Figures and Tables

**Figure 1 cancers-14-01071-f001:**
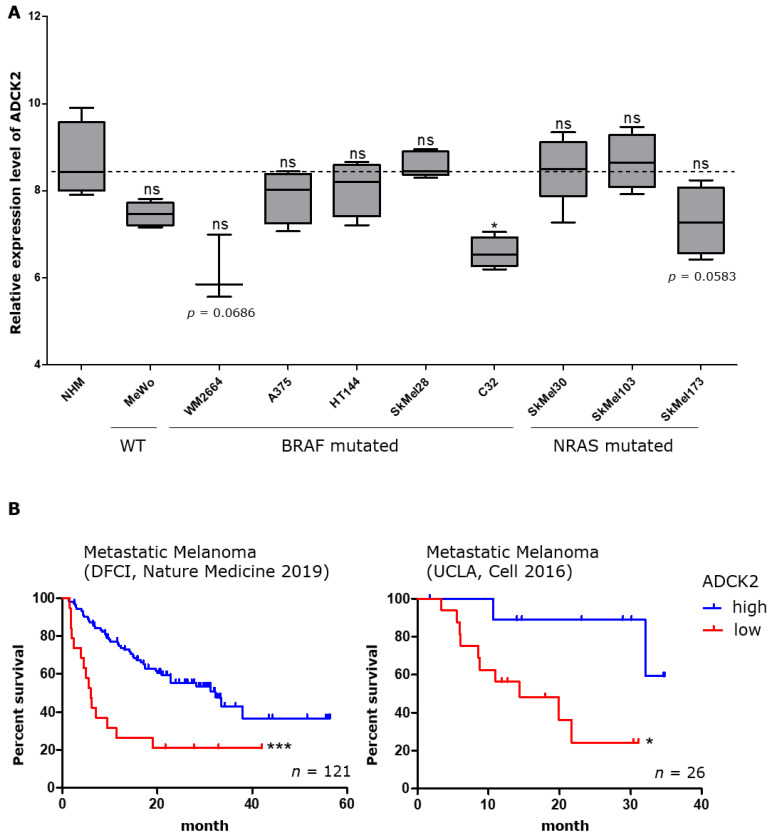
Higher intratumoral levels of ADCK2 correlate with better survival of melanoma patients. (**A**): The relative expression of ADCK2 in nine melanoma cell lines is as high or lower compared to normal human melanocytes and is not affected by the mutational status of the melanoma cells. Statistical analysis was performed with paired *t*-test. (*n* = 4). (**B**): Data from two distinct datasets from the cBioPortal database show that a higher intratumoral level of ADCK2 goes along with better disease-specific survival of melanoma patients. Statistical analysis of the survival data was performed with a log-rank test. Median survival: left graph high = 32.27, low = 5933 months; right graph high = undefined, low = 14.4 months. * *p* ≤ 0.05; *** *p* ≤ 0.001; ns = not significant.

**Figure 2 cancers-14-01071-f002:**
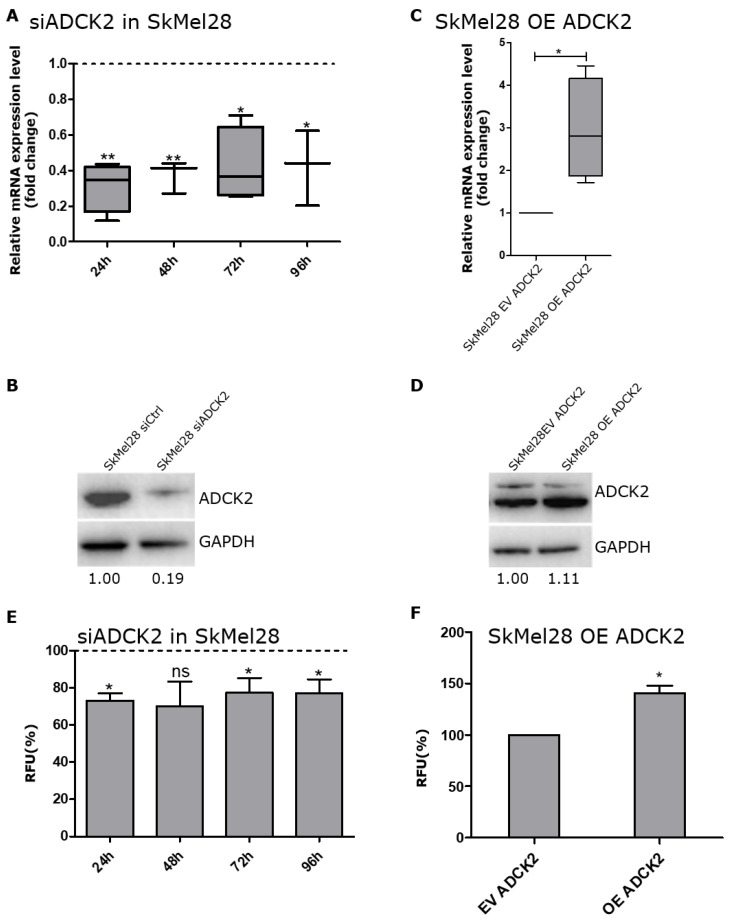
Lower levels of ADCK2 led to lower cell viability. (**A**): The melanoma cell line SkMel28 was transfected with siADCK2, and the cell viability was quantified for several time points after the transfection. The relative mRNA expression level of ADCK2 was reduced as early as 24 h post-transfection and even 96 h after transfection. The mRNA expression was normalized to SkMel28 siControl-transfected cells (*n* = 4). (**B**): Confirmation of a successful knockdown of ADCK2 96 h after transfection via Western blot. (**C**): SkMel28 cells were infected with either an empty vector (EV) control or an ADCK2 overexpression (OE) vector. The relative mRNA expression level increased in the ADCK2 OE cells about 3-fold compared with the EV-transfected cells (*n* = 4). (**D**): Confirmation of the overexpression of ADCK2 in OE ADCK2 cells via Western blot. (**E**): Lower levels of ADCK2 led to a significantly decreased cell viability of SkMel28 cells by about 20% after 24, 48, 72 and 96 h post-transfection compared to cells transfected with siControl (*n* = 6). (**F**): The overexpression of ADCK2 in SkMel28 cells led to a slightly increased cell viability compared to EV SkMel28 cells (time point 24 h) (*n* = 5). All statistical analyses were conducted with paired *t*-tests. * *p* ≤ 0.05; ** *p* ≤ 0.01; ns = not significant; RFU = relative fluorescent units.

**Figure 3 cancers-14-01071-f003:**
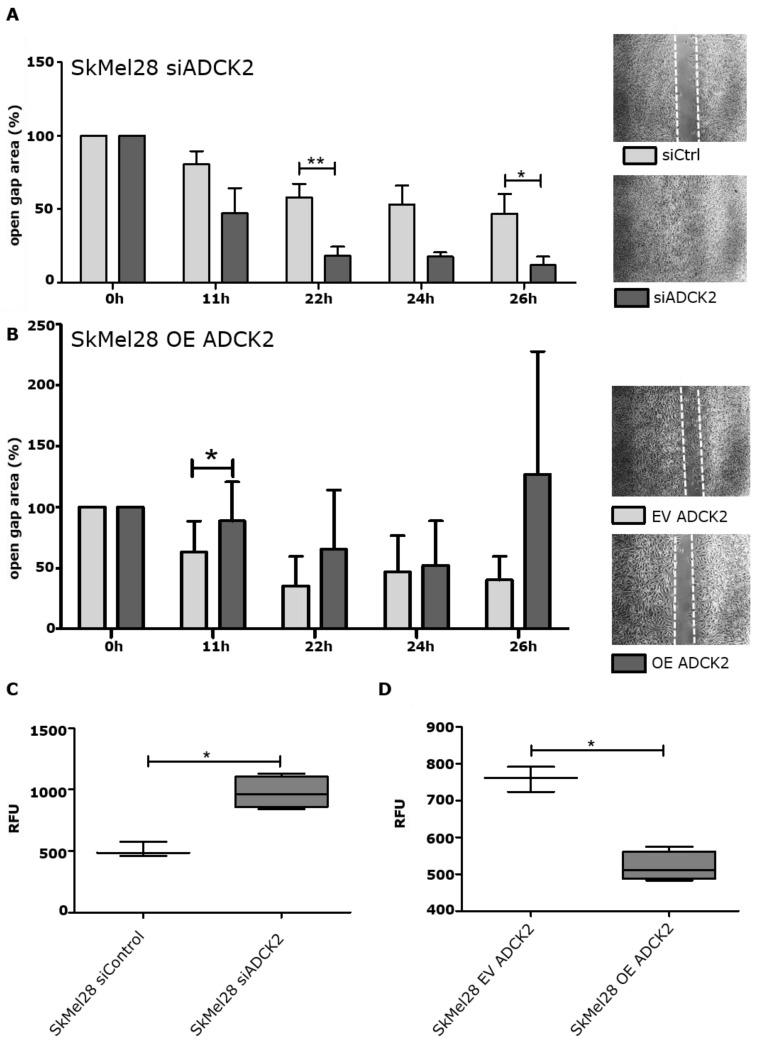
ADCK2 inhibits the migration and invasion of SkMel28 cells. (**A**) Knockdown of ADCK2 led to a faster migration of SkMel28 cells compared to control cells. A significantly faster migration was measured 22 h and 26 h after transfection with siRNAs (*n* = 4). (**B**): ADCK2 overexpression SkMel28 cells migrate slower than EV control cells (*n* = 4). (**C**): Knockdown of ADCK2 in SkMel28 cells leads to a higher invasion capacity through a BME-coated matrix compared to siControl-transfected cells (*n* = 4). (**D**): SkMel28 OE ADCK2 cells show a lower invasion capacity through a BME-coated matrix compared to SkMel28 EV control cells (*n* = 4). All statistical analyses were conducted with paired *t*-test. * *p* ≤ 0.05; ** *p* ≤ 0.01.

**Figure 4 cancers-14-01071-f004:**
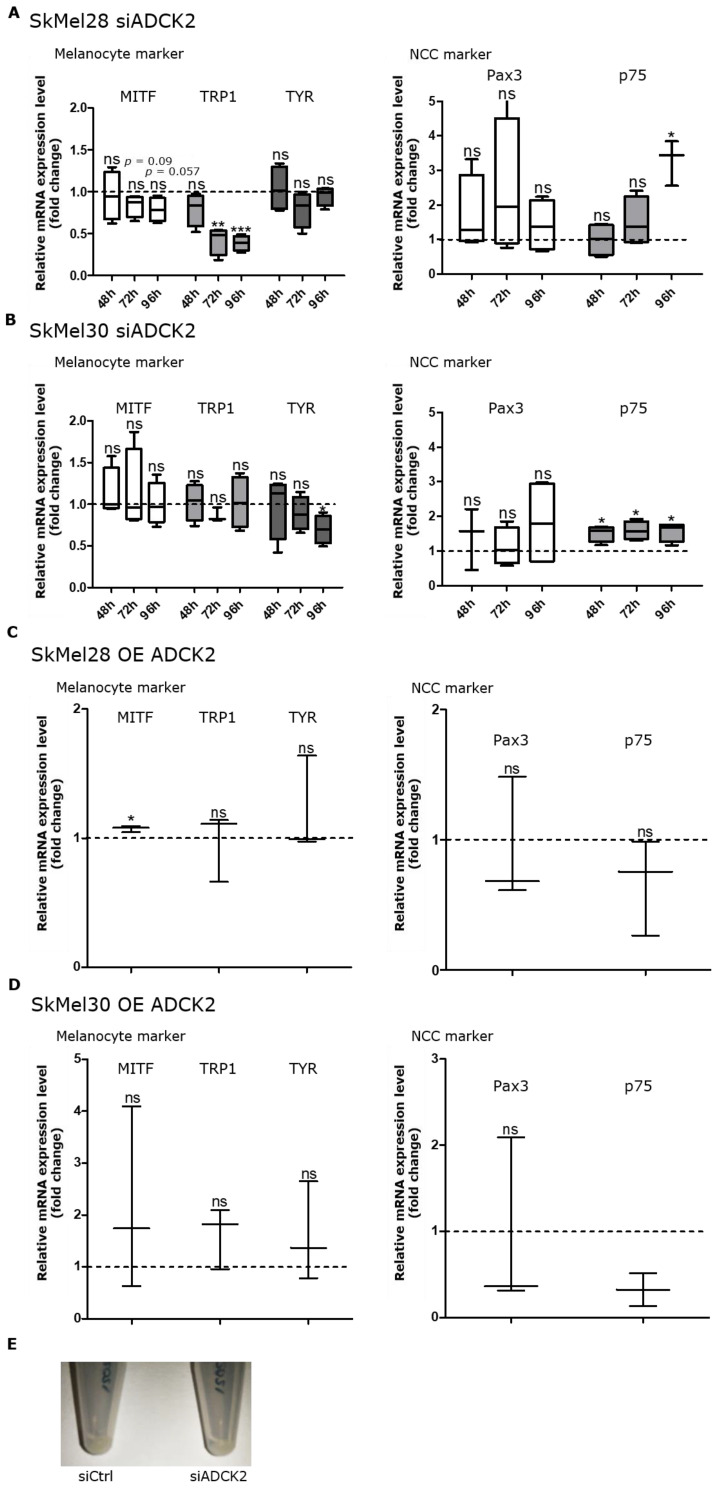
ADCK2 knockdown induces a more dedifferentiated phenotype in melanoma cells. (**A**): SkMel28 siADCK2-transfected cells showed a significantly lower expression of the melanocyte marker TRP1. For TYR and MITF, we could show a tendency for downregulation, which, however, was not significant (left side). The ADCK2 knockdown also led to a higher expression of the NCC marker p75 (right side). (**B**): A knockdown of ADCK2 in SkMel30 cells led to a decreased expression of the melanocyte marker TYR, while the NCC marker p75 was upregulated. All relative mRNA expression levels were normalized to siControl-transfected Skmel28 or SkMel30 cells, respectively (*n* = 4). (**C**): The overexpression of ADCK2 induced upregulation of MITF in SkMel28 cells. The expression of the melanocyte markers TRP1 and TYR, as well as the expression of the NCC markers Pax3 and p75, was not altered. (**D**): The expression of the melanocyte markers MITF, TRP1 and TYR was not changed in SkMel30 ADCK2-overexpressing cells (left side). Additionally, the expression of the NCC markers Pax3 and p75 remained unaltered (*n* = 3). (**E**): The pellets of the pigmented cell line SkMel30 were slightly brighter 96 h after knockdown of ADCK2 compared to control cells. All statistical analyses were conducted with paired *t*-test. * *p* ≤ 0.05; ** *p* ≤ 0.01; *** *p* ≤ 0.001; ns = not significant.

**Figure 5 cancers-14-01071-f005:**
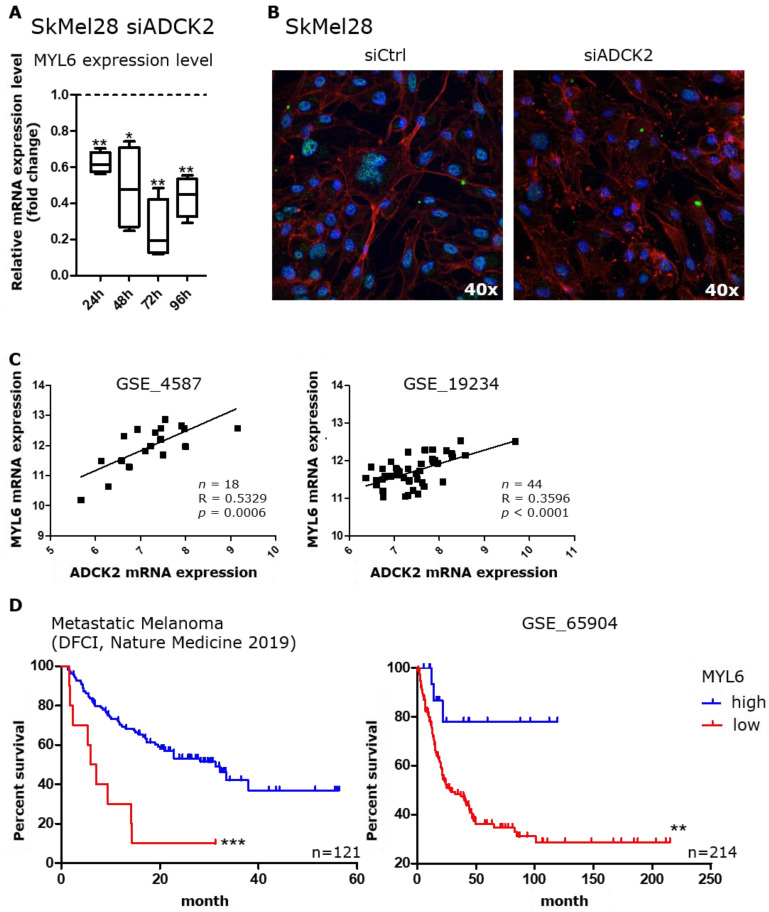
Lower levels of intratumoral MYL6 correlate with a poor outcome for melanoma patients. (**A**): Relative mRNA levels of MYL6 in SkMel28 cells 24, 48, 72 and 96 h after knockdown of ADCK2. MYL6 was significantly downregulated in SkMel28 cells treated with siADCK2 compared to the siControl group. Statistical analysis was conducted with paired *t*-test. *n* = 4. (**B**): Representative images of an immunofluorescence staining of SkMel28 cells 48 h after transfection with either siControl or siADCK2. The amount of MYL6 was greatly reduced after ADCK2 knockdown. Red = Actin; green = MYL6; blue = DAPI. (**C**) The analysis of data from two distinct melanoma datasets from the GEO database revealed a positive correlation between ADCK2 and MYL6 expression. Statistical analysis was performed with Pearson correlation. (**D**): Kaplan–Meier Curves from two distinct datasets from the cBioPortal and the GEO database (GSE_65904) showed that a higher level of intratumoral MYL6 goes along with better disease-specific survival of melanoma patients. Statistical analysis of the survival data was performed with a log-rank test. Median survival: left graph high = 31.27, low = 6517 months; right graph high = undefined, low = 28.3 months. * *p* ≤ 0.05; ** *p* ≤ 0.01; *** *p* ≤ 0.001.

**Figure 6 cancers-14-01071-f006:**
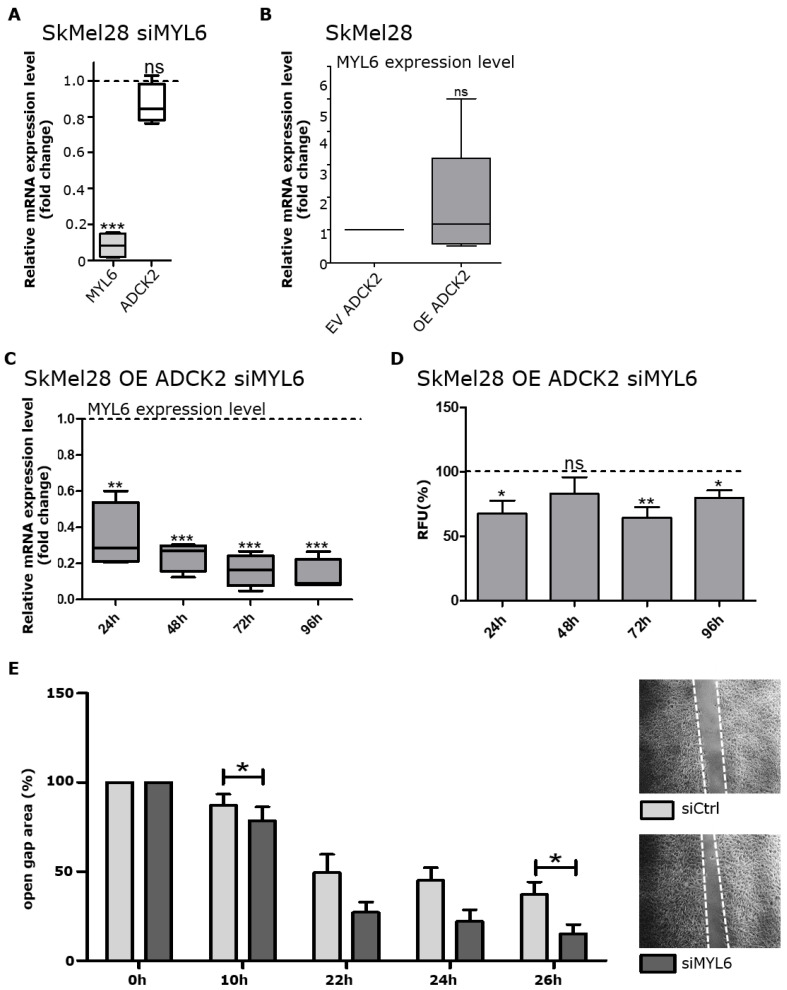
Knockdown of MYL6 negates the effect of ADCK2 OE on melanoma cells. (**A**): The expression of ADCK2 was not altered upon MYL6 knockdown (*n* = 4). (**B**): No significant difference in the expression of MYL6 could be detected between EV and OE ADCK2 cells (*n* = 4). (**C**): MYL6 expression was significantly reduced in SkMel28 OE ADCK2 cells upon treatment with siMYL6 compared to siControl (*n* = 4). (**D**): Lower levels of MYL6 led to a decreased cell viability of SkMel28 OE ADCK2 cells compared to control cells (*n* = 6). (**E**): The migration capacity of SkMel28 OE cells was increased by siMYL6 treatment compared to siControl treatment (*n* = 4). All statistical analyses were conducted with paired *t*-tests. * *p* ≤ 0.05; ** *p* ≤ 0.01; *** *p* ≤ 0.001; ns = not significant.

## Data Availability

We examined the correlation between ADCK2 and MYL6 expression levels in melanoma by analyzing two different GSE datasets (GSE_4587, GSE_19234) from the GEO database (https://www.ncbi.nlm.nih.gov/geo/ (accessed on 12 October 2021). Patient survival data from public patient data sets were obtained from TCGA Pancancer ATLAS database (www.cbioportal.org (accessed on 12 October 2021) and GEO database (https://www.ncbi.nlm.nih.gov/geo/ (accessed on 12 October 2021) (GSE_65904). Kaplan–Meier plots showing patient survival were generated using databases with available survival data.

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
