# Peer review of "ADCK2 Knockdown Affects the Migration of Melanoma Cells via MYL6"

_cancers, 2022, doi:10.3390/cancers14041071_

Round 1

Reviewer 1 Report

-Some formal aspects are missing, such as the meaning of the abbreviations used the first time these items are mentioned (CoQ, ADCK, ER+ve, NCC, NAET…).

-Line 37: a legend explaining the picture should be included.

Introduction:

-Line 55-56: ….upregulation of ADCK2 seems to play a role in tumor survival [14], [15].

The role of ADCK2 could be specified (promote or decrease of tumor survival) as well as the type of cancer, as it is strongly related with the topic.

-Line 56-57:This effect was shown in ER+ve breast cancer cells and GBM cells [8], [16].

What effect? the upregulation of ADCK5 or the ADCK2 role in tumor survival? these sentences could be rewritten.

Methods:

-2.2. siRNA transfection and overexpression:

The quantity of siRNA used is missing.

Results:

-Line 205: 3.1. ADCK2 levels are reduced in melanoma cells compared to melanocytes and patients with high intratumoral ADCK2 expression have a better prognosis

As ADCK2 reduction in melanoma cells seems to be a tendency but it is not clear from the results, I suggest authors remove that from the heading, in a similar way as they did for Figure 1 heading.

I would remove the following sentence from the abstract for the same reason: Line 15 In our study, we saw that the expression of ADCK2 in melanoma cells was lower compared to melanocytes.

-Line 211-212: Furthermore, the expression of ADCK2 in melanoma cells was slightly reduced compared to NHM for most cell lines (Figure 1A).

Authors should mention that this slight reduction is not statistically significant.

-Line 208: (WT: MeWo, BRAF-mutated:...): BRAF and NRAS WT: MeWo.

-Western blot Figures:

I suggest authors normalise the quantitative measurement to 1 or 100 for siControl points so that interpretation of results are clearer.

-The protein name is missing in all Western blot Figures (ADCK2, GAPDH…).

-Ectopic overexpression of ADCK2 increased the mRNA level about 3 fold for SkMel28, MeWo and SkMel30 and about 5-fold for A375 cells compared to an empty vector (EV) control. This overexpression was also confirmed by western blot (Figure 2C, D and Figure S1C, D).

The mRNA expression of ADCK2 in A375, MeWo and SkMel 30 (Fig S1C) cells transfected with the overexpression vector has been compared to each cell line transfected with the empty vector?

-As expected, the cell viability of SkMel28 OE ADCK2, MeWo OE ADCK2 and A375 OE ADCK2 cells was slightly increased after 24, 48, 72 and 96h and after 24h for SkMel30 OE ADCK2 cells (Figure 2F and Figure S2B). 

It seems that only at the earlier time points (24h) of the OE experiments, the effect in viability is statistically significant. This should be mentioned in the text. 

-Figure 3 legend : (c): SkMel28 OE ADCK2 cells show a lower invasion capacity through a BME coated matrix compared to SkMel28 EV control cells. n=4 (d): Knockdown of ADCK2 in SkMel28 cells leads to a higher invasion capacity through a BME coated matrix compared to siControl transfected cells.

Figure 3C is the knockdown, while 3D is the overexpression experiment.

-Fig 4C: the pellet of the empty vector (EV) should be included in the photo in order to make a proper comparison with the OE pellet.

-Figure 6. The empty vector EV condition should be included in the MYL6 knocking down experiments to properly compare the abrogating effects of silencing MYL6 in ADCK2 OE cells. At least with SKMel28 cells.

-Figure 6. It would be interesting to add as a control the expression of  ADCK2 when knocking down MYL6.

-Fig S6 D: in order to properly compare the pigmentation effects mentioned in the text, a photo of the pellets from skmel30 EV (empty vector) transfected with siControl and siMYL6 should be included.

-Line 363: With these experiments, we confirmed a direct functional connection between ADCK2 and MYL6.

I don´t think a direct connection has been confirmed with the experiments shown. The sentence should be rewritten.

Supplementary:

-Figure legends in Supplementary material are missing/should be more detailed.

-Table S1: Sequences of the primers used for qPCR. Table S1 is missing in the supplementary material.

Author Response

Reviewer 1

-Some formal aspects are missing, such as the meaning of the abbreviations used the first time these items are mentioned (CoQ, ADCK, ER+ve, NCC, NAET…).

Answer: Thank you for this information. We have added the meanings to the abbreviations.

-Line 37: a legend explaining the picture should be included.

Answer: Thank you for this advice. We included a legend.

Introduction:

-Line 55-56: ….upregulation of ADCK2 seems to play a role in tumor survival [14], [15].

The role of ADCK2 could be specified (promote or decrease of tumor survival) as well as the type of cancer, as it is strongly related with the topic.

Answer: Thank you for this comment. The role of ADCK2 and the type of cancer were specified.

-Line 56-57:This effect was shown in ER+ve breast cancer cells and GBM cells [8], [16].

What effect? the upregulation of ADCK5 or the ADCK2 role in tumor survival? these sentences could be rewritten.

Answer: Thank you for this hint. We rewrote the sentence to clarify the statement.

Methods:

-2.2. siRNA transfection and overexpression:

The quantity of siRNA used is missing.

Answer: Thank you for this comment. We added the quantity of siRNA to the respective section.

Results:

-Line 205: 3.1. ADCK2 levels are reduced in melanoma cells compared to melanocytes and patients with high intratumoral ADCK2 expression have a better prognosis

As ADCK2 reduction in melanoma cells seems to be a tendency but it is not clear from the results, I suggest authors remove that from the heading, in a similar way as they did for Figure 1 heading.

I would remove the following sentence from the abstract for the same reason: Line 15 In our study, we saw that the expression of ADCK2 in melanoma cells was lower compared to melanocytes.

Answer: Thank you for pointing out this weak point. The wording that „ADCK2 expression was reduced in melanoma cells compared to NHM“ was removed from the abstract and the heading of paragraph 3.1.

-Line 211-212: Furthermore, the expression of ADCK2 in melanoma cells was slightly reduced compared to NHM for most cell lines (Figure 1A).

Authors should mention that this slight reduction is not statistically significant.

Answer: Thank you for this important advice. We now wrote that there is no statistical significance.

-Line 208: (WT: MeWo, BRAF-mutated:...): BRAF and NRAS WT: MeWo.

Answer: Thank you for this helpful comment. We changed it accordingly.

-Western blot Figures:

I suggest authors normalise the quantitative measurement to 1 or 100 for siControl points so that interpretation of results are clearer.

Answer: Thank you for this helpful suggestion. We calculated it accordingly and changed it for all Western Blot figures.

-The protein name is missing in all Western blot Figures (ADCK2, GAPDH…).

Answer: Thank you for pointing out this issue. We now added the protein names to all Western Blot figures.

-Ectopic overexpression of ADCK2 increased the mRNA level about 3 fold for SkMel28, MeWo and SkMel30 and about 5-fold for A375 cells compared to an empty vector (EV) control. This overexpression was also confirmed by western blot (Figure 2C, D and Figure S1C, D).

The mRNA expression of ADCK2 in A375, MeWo and SkMel 30 (Fig S1C) cells transfected with the overexpression vector has been compared to each cell line transfected with the empty vector?

Answer: Thank you for this comment. Yes, we compared the ADCK2-overexpressing cell lines to the control that was transfected with an empty vector construct.

-As expected, the cell viability of SkMel28 OE ADCK2, MeWo OE ADCK2 and A375 OE ADCK2 cells was slightly increased after 24, 48, 72 and 96h and after 24h for SkMel30 OE ADCK2 cells (Figure 2F and Figure S2B). 

It seems that only at the earlier time points (24h) of the OE experiments, the effect in viability is statistically significant. This should be mentioned in the text. 

Answer: Thank you for this comment. We thoroughly reanalyzed the cell viability data and came to the conclusion that the effect of ADCK2 on cell viability is probably low or negligible. Due to the rapid growth of the SkMel28 cells, later time points than 24h did not yield any reliable data. However, our reanalysis also showed that neither MeWo, A375 nor SkMel30 showed a difference in cell viability upon ADCK2 overexpression. We changed Figure 2F as well as S2B accordingly.

-Figure 3 legend : (c): SkMel28 OE ADCK2 cells show a lower invasion capacity through a BME coated matrix compared to SkMel28 EV control cells. n=4 (d): Knockdown of ADCK2 in SkMel28 cells leads to a higher invasion capacity through a BME coated matrix compared to siControl transfected cells.

Figure 3C is the knockdown, while 3D is the overexpression experiment.

Answer: Thank you for this comment. We corrected this mistake.

-Fig 4C: the pellet of the empty vector (EV) should be included in the photo in order to make a proper comparison with the OE pellet.

Answer: Thank you for this comment. Since there was no significant difference between the melanocyte and NCC marker expression on RNA level upon ADCK2 overxpression we excluded this figure and changed the legend accordingly.

-Figure 6. The empty vector EV condition should be included in the MYL6 knocking down experiments to properly compare the abrogating effects of silencing MYL6 in ADCK2 OE cells. At least with SKMel28 cells.

Answer: Thank you for this comment. We could show that the MYL6 expression did not change upon overexpressing ADCK2 and we included the respective graph in Figure 6. For this reason, we only used the OE ADCK2 cell lines and as a reference the siControl transfected cells for all siMYL6 experiments.

-Figure 6. It would be interesting to add as a control the expression of ADCK2 when knocking down MYL6.

Answer: We performed an additional experiment in which we knocked down MYL6 in parental SkMel28 cells and checked the ADCK2 expression. We included these results into Figure 6 and also in the text. Additionally, we changed the title of our study in order to express that the connection between ADCK2 and MYL6 is rather indirect, since the overexpression of ADCK2 does not affect the expression of MYL6.

-Fig S6 D: in order to properly compare the pigmentation effects mentioned in the text, a photo of the pellets from skmel30 EV (empty vector) transfected with siControl and siMYL6 should be included.

Answer: Thank you for this comment. As explained above we did not see a difference in MYL6 expression between ADCK2 OE and EV control cells and thus did not perform the knockdown experiment with the EV control cells.

-Line 363: With these experiments, we confirmed a direct functional connection between ADCK2 and MYL6.

I don´t think a direct connection has been confirmed with the experiments shown. The sentence should be rewritten.

Answer: Thank you for this comment. We removed the word „direct“ from the sentence.

Supplementary:

-Figure legends in Supplementary material are missing/should be more detailed.

Answer: Thank you for this comment. The Figure legends were not missing. There must have been a problem with the processing of the uploaded document.

-Table S1: Sequences of the primers used for qPCR. Table S1 is missing in the supplementary material.

Answer: Thank you for this comment. Table S1 was actually included in the supplementary material. There must have been a problem with the processing of the uploaded document.

Reviewer 2 Report

The Authors have investigated the role of ADCK2 on melanoma cell motility and differentiation status upon siRNA-mediated knockdown or stable overexpression of ADCK2. The topic is interesting and relative novel as the function of ADCK2 in melanoma is elusive. The conclusions are generally consistent with the evidence provided, and all presented data are technically valid. Appropriate references have been refered to. Figures are clearly presented (please see comment below).

Specific comments:

  1. Please write protein names in the Fig. 2B and D for clarity. In addition, clarify in figure legend or M&M what the values below the blots mean.
  2. Gene expression should be studied and shown in Fig. 4 for overexpression of ADCK2.

Author Response

Reviewer 2:

The Authors have investigated the role of ADCK2 on melanoma cell motility and differentiation status upon siRNA-mediated knockdown or stable overexpression of ADCK2. The topic is interesting and relative novel as the function of ADCK2 in melanoma is elusive. The conclusions are generally consistent with the evidence provided, and all presented data are technically valid. Appropriate references have been refered to. Figures are clearly presented (please see comment below).

Specific comments:

  1. Please write protein names in the Fig. 2B and D for clarity. In addition, clarify in figure legend or M&M what the values below the blots mean.

Answer: Thank you for this helpful comment. We included all protein names to all western blot figures and added in the M&M section that the values describe the quantitative measurement of the bands normalized to the respective control.

  1. Gene expression should be studied and shown in Fig. 4 for overexpression of ADCK2.

Answer: Thank you for this nice suggestion. We studied the gene expression of NCC and melanocyte markers in OE ADCK2 cells and included our results now in Fig. 4.

Reviewer 3 Report

Although early melanoma is curable, metastatic melanoma is highly fatal due to metastasis and resistance to treatments. Therefore, understanding the migration of melanoma cells is critical to develop improved treatment. Work by Vierthaler et al. investigated the regulatory effects of ADCK2 on the migration and proliferation of melanoma cells. The authors found that the levels of ADCK2 in melanoma cells are relatively lower than melanocytes. In addition, they observed that patients with lower ADCK2 had significantly increased mortality. Mechanistically, ADCK2 promotes the proliferation, while suppresses the migration of melanoma cells in a MYL6-dependnent manner. However, I have several concerns which are listed below:

Major:

  1. The authors evaluated the expression levels of ADCK2 in various melanoma cell lines (Figure 1A). However, most of the studies were performed using SkMel28 cells. How SkMel28 was selected over other cell lines?

  1. I did not see any difference in migration following ADCK2 overexpression in MeWo (Figure S3B). Therefore, the conclusion in line 268 is inappropriate.

  1. The in vitro experiments were well-designed. However, the main weakness of this paper is lack of in vivo investigation.

Minor:

  1. The complete term of NCC (neural crest cells) should be defined at the first appearance.

Author Response

Reviewer 3:

Although early melanoma is curable, metastatic melanoma is highly fatal due to metastasis and resistance to treatments. Therefore, understanding the migration of melanoma cells is critical to develop improved treatment. Work by Vierthaler et al. investigated the regulatory effects of ADCK2 on the migration and proliferation of melanoma cells. The authors found that the levels of ADCK2 in melanoma cells are relatively lower than melanocytes. In addition, they observed that patients with lower ADCK2 had significantly increased mortality. Mechanistically, ADCK2 promotes the proliferation, while suppresses the migration of melanoma cells in a MYL6-dependnent manner. However, I have several concerns which are listed below:

Major:

  1. The authors evaluated the expression levels of ADCK2 in various melanoma cell lines (Figure 1A). However, most of the studies were performed using SkMel28 cells. How SkMel28 was selected over other cell lines?

Answer: Thank you for this comment. We chose SKMel28 cells because they represent the BRAF-mutated cell line with the highest endogenous expression of ADCK2.

  1. I did not see any difference in migration following ADCK2 overexpression in MeWo (Figure S3B). Therefore, the conclusion in line 268 is inappropriate.

Answer: Thank you for this comment. We removed this sentence.

  1. The in vitro experiments were well-designed. However, the main weakness of this paper is lack of in vivo investigation.

Answer: Thank you for this comment. Unfortunately, the extremely strict German restrictions on animal experiments result in very long approval processes for in vivo experiments. Due to time constraints we could not include in vivo experiments into this study. However, in vivo experiments will definitely be considered for a follow-up study on this interesting topic.

Minor:

  1. The complete term of NCC (neural crest cells) should be defined at the first appearance.

Answer: Thank you for this comment. The complete term was added.

Round 2

Reviewer 2 Report

All comments have been addressed.

Reviewer 3 Report

The authors have addressed my concerns and questions.